# Aspirin Caused Intestinal Damage through FXR and ET-1 Signaling Pathways

**DOI:** 10.3390/ijms25063424

**Published:** 2024-03-18

**Authors:** Qiuxia Lin, Binbin Zhang, Manyun Dai, Yan Cheng, Fei Li

**Affiliations:** 1Laboratory of Hepatointestinal Diseases and Metabolism, State Key Laboratory of Biotherapy, West China Hospital, Sichuan University, Chengdu 610041, China; 2021224070025@stu.scu.edu.cn (Q.L.); 2021324020167@stu.scu.edu.cn (B.Z.); chengyan@jxutcm.edu.cn (Y.C.); 2Department of Gastroenterology & Hepatology, Frontiers Science Center for Disease-Related Molecular Network, West China Hospital, Sichuan University, Chengdu 610041, China; 3State Key Laboratory of Respiratory Health and Multimorbidity, West China Hospital, Sichuan University, Chengdu 610041, China

**Keywords:** aspirin, intestinal injury, FXR, endothelin 1, bile acid

## Abstract

Aspirin is a non-steroidal, anti-inflammatory drug often used long term. However, long-term or large doses will cause gastrointestinal adverse reactions. To explore the mechanism of intestinal damage, we used non-targeted metabolomics; farnesoid X receptor (FXR) knockout mice, which were compared with wild-type mice; FXR agonists obeticholic acid (OCA) and chenodeoxycholic acid (CDCA); and endothelin-producing inhibitor estradiol to explore the mechanisms of acute and chronic intestinal injuries induced by aspirin from the perspective of molecular biology. Changes were found in the bile acids taurocholate acid (TCA) and tauro-β-muricholic acid (T-β-MCA) in the duodenum, and we detected a significant inhibition of FXR target genes. After additional administration of the FXR agonists OCA and CDCA, duodenal villus damage and inflammation were effectively improved. The results in the FXR knockout mice and wild-type mice showed that the overexpression of endothelin 1 (ET-1) was independent of FXR regulation after aspirin exposure, whereas CDCA was able to restore the activation of ET-1, which was induced by aspirin in wild-type mice in an FXR-dependent manner. The inhibition of ET-1 production could also effectively protect against small bowel damage. Therefore, the study revealed the key roles of the FXR and ET-1 pathways in acute and chronic aspirin-induced intestinal injuries, as well as strategies on alleviating aspirin-induced gastrointestinal injury by activating FXR and inhibiting ET-1 overexpression.

## 1. Introduction

Aspirin is a generally used non-steroidal, anti-inflammatory drug (NSAID), which shows prominent analgesic, antipyretic, and anti-inflammatory effects. It is a commonly used drug for the treatment of rheumatoid arthritis [1,2]. Meanwhile, due to a unique effect on the impeding platelet aggregation, aspirin is also used as reliable medicine for preventing cardiovascular and cerebrovascular diseases [3]. However, the side effects restrict the applications of aspirin, especially for damage of the digestive tract [4]. Evidence suggests that a large dose or long-term treatment with aspirin could induce a gastrointestinal mucosa injury, even after a pretreatment with prostaglandin analogues [5]. Although researchers believe that low-dose aspirin is safe for the digestive tract in most cases, long-term low-dose uses of aspirin still carry an increased risk of gastrointestinal bleeding in patients [6]. Low-dose administration prominently increases the risk of gastrointestinal ulcers and severe bleeding by 2–4 times, especially in the elderly [7]. A gastroduodenal ulcer was found in 11% of the endoscopic reports of patients treated with low-dose aspirin (≤325 mg/d) [8]. The combination of a proton pump inhibitor, such as omeprazole, or aspirin enteric-coated tablets can effectively improve upper gastrointestinal adverse reactions caused by aspirin, yet there is a risk of aggravating lower gastrointestinal injury [9].

Previous studies show that the partial mechanism of a digestive tract injury induced by aspirin mainly focuses on the upper digestive tract, especially in the stomach [10]. Stomach damage in aspirin users is thought to be related to COX enzyme inhibition, adrenaline production, gastric acid secretion, and bleeding [11,12,13]. Nevertheless, gastrointestinal damage was not found in COX1 knockout mice [14], so the mechanism of enteric canal adverse reaction still needs further research. Recent observational epidemiological evidence has raised concerns that aspirin discontinuation due to treatment noncompliance, surgery, or side effects may increase the risk of thrombosis [15]. Simultaneously, the long-term use of aspirin would increase the risk of a lower digestive tract hemorrhage and is generally found to be more serious [16]. Therefore, it is necessary to explore the mechanism of small bowel injury caused by aspirin.

Bile acid is involved in the digestion and absorption of lipids in food, regulates cholesterol and glucose metabolism, and plays a momentous role in human digestion, absorption, and metabolism [17]. FXR is a nuclear receptor mainly expressed in tissues such as the liver and intestine and is a specific receptor that functions to balance bile acid metabolism to maintain homeostasis in the body. It is also the cornerstone of bile acid regulation in cell metabolism [18]. In the liver, FXR activation lowers the mRNA levels of inflammatory genes and curbs the production of inflammatory factors, thereby decelerating liver inflammation and fibrosis [19]. In the intestine, FXR activation can impede the activation of inflammatory signaling pathways, foster the expression of intestinal tight junction proteins (claudin-1 and occludin), and minimize the permeability of the intestinal barrier, thus suppressing intestinal inflammation and barrier damage [20,21]. Recent research has linked gut FXR to intestinal bleeding, and animals are more susceptible to drugs in the absence of FXR. FXR–JNK axis was dysregulated in the gastrointestinal tract when intestinal injury and bleeding occurred [22]. Aspirin can cause more severe gastric mucosal damage in *Fxr*-null (*Fxr*^−/−^) mice, and the activation of FXR can improve NSAID damage to the stomach and duodenum; however, its mechanism is still not clear [23].

In the study, the effects of FXR agonists were observed in wild-type mice and *Fxr*-null (*Fxr*^−/−^) mice. According to the results from the models of acute and chronic doses of aspirin in mice, it was revealed that aspirin induces intestinal injury through its effect on the FXR and ET-1 pathways, thereby activating JNK and inflammation.

## 2. Results

### 2.1. Non-Targeted Metabolomics Identified the Disruption of Bile Acid Pools and Inhibition of FXR in Aspirin-Induced Intestinal Injury

In the study, the oral administration of aspirin at 300 mg/kg caused duodenal lesions 3 h after medication. H&E staining showed severe damage to the villous morphology, showing shortening, swelling, and the formation of vacuoles at the apex (Figure 1A). Furthermore, the integrity of the villi was compromised. The activity of DAO in the blood was significantly elevated in the model group (Figure 1B), and an increase in Evans Blue dye content was observed in the intestinal contents after intravenous injection (Figure 1C and Appendix B Figure A1A), indicating the disruption of the intestinal mechanical barrier. A fecal occult blood test kit measured the hemoglobin content in the cecum and found that the administration of aspirin caused gastrointestinal bleeding in mice (Figure A1B). In addition, the up-regulation of the *Gpx4* gene and the decrease in the GSH/GSSG ratio suggested that aspirin also induced oxidative stress in cells (Figure 1B,C).

The expression levels of the inflammatory factors (*Il6*, *Ccl2*, *Cox2*, *Il1b*, and *Tnfa*) and genes of the JNK pathway (*Caspase3*, *Cfos*, *Cjun*, *Hsf1*, and *p53*) were measured in the duodenum. The mRNA levels of these genes were remarkably up-regulated in the model group (Figure 1D,E), promoting the activation of apoptosis and inflammation in the duodenal cells. We also found a significant increase in ET-1 gene expression in the aspirin-induced enteritis model (Figure 1E).

To elucidate the mechanism of aspirin-induced duodenal damage, untargeted metabolomics were employed to analyze the metabolites that may contribute to the disruption of the normal physiological function in the target intestinal segment. The PCA results show a clear differentiation between the model and control groups after the administration of aspirin (Figure 1F). The results of OPLS-DA showed that the most significantly changed metabolites were conjugated bile acids (TCA and T-β-MCA), and some kinds of free bile acids were markedly reduced (CA and CDCA) (Figure 1G,H and Appendix B Figure A1C), indicating a severe disruption in the bile acid metabolism of the duodenal segment under aspirin intervention. In the duodenum, ASA markedly reduced the mRNA levels of the FXR target genes (*Shp*, *Ostb*, and *Ibabp*) (Figure 1I), suggesting that aspirin-induced duodenal damage may be related to the disruption of the bile acid composition and inhibition of FXR.

### 2.2. Activation of FXR Can Significantly Reduce Duodenum Injury Induced by Aspirin and Alleviate ET-1 Overexpression

OCA is an agonist of FXR and inhibits the expression of the cytochrome 7A1 (CYP7A1) by activating FXR. Since CYP7A1 is the rate-limiting enzyme in bile acid synthesis, OCA can inhibit bile acid synthesis and is used to treat primary biliary cholangitis and non-alcoholic fatty liver disease [24]. To verify whether the inhibition of FXR is directly associated with the induction of inflammation and oxidative stress, CDCA (15 mg/kg) or OCA (10 mg/kg) [25] was administered 5 days in advance.

Both CDCA and OCA effectively restored the pathological damage to the duodenal villi, increased the villus length, reduced edema and vacuoles, maintained the normal morphology of the villi (Figure 2A and Appendix B Figure A2A), and reduced the levels of DAO in the blood to restore the normal barrier function of the intestine (Figure 2B). In addition, OCA reduced the MDA levels in the duodenum after drug exposure and relieved digestive tract injury (Figure A2B–D). Both of them were effective in alleviating the decrease in FXR protein levels in the duodenums of mice treated with aspirin, activating the FXR pathway (Figure 2C–E and Appendix B Figure A2F,G). CDCA restored the stability of the bile acid pool in the body, lowered the levels of conjugated bile acids, and compensated for the decrease in CDCA levels (Figure 2F).

Additionally, CDCA effectively alleviated the up-regulation of inflammatory factors (*Cox2*, *Ccl2*, *Il6*, *Tnfa*, and *Il1b*) and JNK pathway genes in the small intestine at the gene level (Figure 3A,B), while reducing the transcription levels of iNOS and ET-1 (Figure 3C, D), exerting effective anti-inflammatory and antioxidant effects. The same results were found in mice given OCA (Figure A2E). The Western blot results show that the activation of FXR prevented the activation of the JNK pathway caused by aspirin, lowering the phosphorylation levels of *Cfos*, *Stat3*, and JNK, and restoring the normal physiological function of the JNK pathway in cells (Figure 3E,F and Appendix B Figure A2G,H).

### 2.3. The Role of FXR in Regulating ET-1 Expression Was Verified in Fxr-Knockout Mice

To verify the role of FXR, aspirin was administered at 300 mg/kg to mice with systemic FXR knockout (Figure A4A), and a protective group was established by administering CDCA at a dose of 15 mg/kg in advance.

The results show that both the ASA group and the CDCA pre-treated group exhibited prominent intestinal damage (Figure 4A–H). H&E staining showed the compromised integrity of the duodenal villi, edema, and shortened and thickened villi along with cystic vacuolation (Figure 4A). The levels of DAO in the sera were not different between the aspirin group and the CDCA group (Figure 4B). Compared with the control group, the mRNA levels of iNOS, inflammatory factors (*Cox2*, *Ccl2*, *Il6*, and *Tnfa*), and JNK pathway genes (*Cfos*, *Cjun*, and *Caspase3*) were observed to distinctly increase in both the ASA and ASA + CDCA groups, and there were no significant differences between them (Figure 4C,E,F). Moreover, the phosphorylation levels of JNK and *Stat3* proteins showed the same elevated trend in all the aspirin-exposed group (Figure 4G,H). It was found that ET-1 expression remained consistently high in FXR knockout mice after aspirin administration, even with a CDCA pretreatment (Figure 4D,G,H).

### 2.4. Inhibition of ET-1 Improves Inflammation and Oxidative Stress in Intestinal Injury

To investigate the role of ET-1 and its relationship with FXR in aspirin-induced duodenal injury, mice were intraperitoneally pre-injected with the ET-1 inhibitor estradiol before aspirin was given.

The villous damage and increased DAO levels observed in the aspirin group were effectively restored in mice pre-treated with estradiol (Figure 5A and Appendix B Figure A4B). Additionally, estradiol effectively reduced the phosphorylation of *Cfos* and *Stat3* in the JNK pathway, as well as the transcription levels of inflammatory factors (*Cox2*, *Ccl2*, *Il6*, *Tnfa*, and *Il1b*) in the aspirin-induced duodenum by inhibiting ET-1 transcription (Figure 5B–D,G,H). Meanwhile, the expression of iNOS was effectually lessened with estradiol injection (Figure 5E). These findings indicated that the up-regulation of ET-1 in this model was probably related to JNK and inflammatory signaling pathways, and the inhibited ET-1 overexpression effectively reduced inflammation and injury. We used qPCR to measure the levels of FXR signaling pathway target genes in the duodenum to better explain the regulation between ET-1 overexpression and FXR in this intestinal injury model. The results show that although estradiol improved oxidative stress and inflammatory damage caused by FXR inhibition, it had no effect on the FXR inhibition induced by aspirin (Figure 5F–H).

### 2.5. CDCA Improved Aspirin-Induced Chronic Duodenal and Ileum Damage through FXR and ET-1 Pathways

To better simulate the clinical long-term, low-dose use of aspirin, a dosage of 150 mg/kg was orally administered to mice for 14 days (Figure A4C) [26]. In the chronic mouse model, we observed significant villus damage, increased DAO levels, the inhibition of the FXR pathway, the activation of JNK, inflammatory signals, and oxidative stress, and after CDCA was administered for 7 days, all these injuries were improved (Figure 6A–G). Moreover, we measured the levels of inflammation and oxidative stress-related genes in the ileum to observe the effects of aspirin on different segments of the intestine. Surprisingly, in the acute model, the ileum segment did not show obvious damage, and there were no changes in the transcription levels of JNK, inflammation, or oxidative stress (Figure A1D,E). Simultaneously, short-term aspirin did not cause a low expression of FXR target genes in the ileum (Figure A1F). Nevertheless, in the chronic model, we observed an increase in the levels of *Cfos*, *Caspase3*, and inflammatory factors (*Cox2*, *Ccl2*, *Il6*, and *Il1b*) in the ileal tissue, as well as a significant up-regulation in the ET-1 gene levels (Figure 6H–J). To confirm whether the damage caused by low-dose aspirin in the ileum segment was similar to that in the duodenum segment, several target genes of FXR were measured. It was found that *Shp*, *Fgf15*, and *Ibabp* were significantly inhibited, indicating that the FXR pathway in the ileum was also disturbed (Figure 6K).

At the same time, we also detected the expression of the FXR protein in duodenal and ileal tissues and found that ASA decreased FXR expression in the ileum. CDCA significantly activated FXR in the duodenum and ileum (Figure A4D,E).

## 3. Discussion

In this study, we identified the disturbance of bile acids in the duodenum after aspirin administration using non-targeted metabolomics. The ratio of conjugated bile acids to unconjugated bile acids was remarkably increased. In the aspirin-induced intestinal injury model, conjugated bile acids (T-β-MCA, TCA, TUDCA, TDCA, and TLCA) were significantly increased, while CA and CDCA were distinctly decreased. Previous studies have suggested that the accumulation of conjugated bile acids in the gut may directly down-regulate the FXR–FGF15 signaling pathway in the gut [27], which meant that the disordered bile acid pool might cause the inhibition of FXR in an aspirin-induced intestinal injury. When CDCA was given to the mice, it significantly restored the inhibition of FXR and intestinal damage. However, duodenal damage in Fxr-null mice was not attenuated by CDCA, which indicated that CDCA improved intestinal damage through a FXR-dependent way, and FXR played a key role in the intestinal damage caused by aspirin. Studies have demonstrated the protective effect of CDCA on aspirin-induced gastric mucosa injury and naproxen-induced duodenal injury, and non-steroidal anti-inflammatory drugs tended to cause more severe stomach damage in Fxr-null mice, which could not be mitigated by CDCA [23]. We found that CDCA and OCA could also reduce the small intestine injury caused by aspirin, further proving that the activation of FXR is a very effective way to alleviate the gastrointestinal adverse reactions of NSAIDs.

ET-1 is a peptide consisting of 21 amino acids and is mainly secreted by endothelial cells [28]. It has a strong vasoconstrictive effect and plays an important role in maintaining the basal vascular tone and cardiovascular homeostasis [29,30]. Researchers have found a close relationship between ET-1 and intestinal inflammation, which is a chronic inflammatory bowel disease that includes ulcerative colitis and Crohn’s disease [31]. Studies have shown significantly elevated levels of ET-1 in the plasma and tissues of patients with intestinal inflammation, so it is suggested to be a marker of endothelial dysfunction; increased levels of ET-1 in the blood of patients with colitis support this idea [32]. ET-1 is involved in the occurrence and development of intestinal inflammation by affecting blood flow, inflammatory responses, and intestinal epithelial cell function [32,33,34]. In fact, in addition to affecting the endothelial function, ET-1 G-protein-coupled receptors exist on the cell membranes of many tissues in the body [35,36,37]. Additionally, the overexpression of ET-1 is related to the phosphorylation level of JNK [38], which induces the phosphorylation of JNK and c-Jun in time-dependent manners, and the inhibition of JNK can negatively regulate the ET-1 transcription level as well [39]. Meanwhile, ET-1 stimulated the activation of c-Jun/activator protein 1 (AP-1) through the Gq/i protein-coupled ETB receptor, then bonded and activated the COX2 promoter, upregulated the transcription level of COX2, and induced inflammation of cerebral microvascular endothelial cells [40]. We found that the normal expression of FXR was a necessary precursor for the regulation of ET-1 in aspirin-induced small intestine injury, because ET-1 activation can be restored by FXR agonists, although it was activated after aspirin exposure in Fxr-null mice. Previous studies showed that FXR can indirectly regulate the transcription level of ET-1 by regulating the level of its transcription factor, AP-1 [41]. Our study further indicated that activation of FXR can negatively regulate ET-1 expression in ASA induced intestinal injury. However, it was not clear whether the overexpression of ET-1 was directly due to FXR inhibition, whereas the CDCA in this model improved the abnormal activation of ET-1 through an FXR-dependent way, which meant that ET-1 was partly regulated by FXR. The expression level of ET-1 increased by 3.1 times in indomethacin gastric mucosal injury [42], and the correlation between ET-1 overexpression and JNK and inflammatory activation after aspirin administration has been further confirmed in our results, suggesting the reliability of ET-1 as a reference factor for the existence of NSAID-induced intestinal injuries. Few researchers have used ET-1 as a direct target to alleviate gastrointestinal mucosal injury. Our study indicated that ET-1 may be a potential therapeutic target for gastrointestinal adverse reactions of NSAIDs, and its expression could be regulated by directly inhibiting or activating FXR to achieve prevention and treatment purposes.

Clinical experience shows that the long-term use of low-dose aspirin is prone to cause adverse reactions in the lower digestive tract [43], and the chronic model we constructed also reflects this feature. There are obvious injuries in both the duodenum and ileum, and the injury mechanisms of the two parts of intestinal tissues are similar from the perspective of genetic changes. FXR and ET-1 induce JNK and inflammatory activation and showed the same profile in all the different intestinal lesions. It is worth noting that in the acute injury, the duodenal segment was extremely damaged, and the levels of inflammation and oxidative stress were significantly increased, while the ileum segment did not produce any abnormal changes. However, the inhibitory effect of FXR was widespread after aspirin administration, which may be related to the mode of administration.

CDCA is an endogenous ligand of FXR, which can activate the FXR receptor and promote its nuclear translocation to bind to reaction elements, thereby regulating gene transcription. Our results suggest that FXR protein levels are significantly up-regulated after CDCA and OCA administration, which may be related to the post-translational modification of FXR. Research revealed that the ubiquitination/de-ubiquitination process of FXR is very important for the activation of the FXR ligand, and the ubiquitination level of FXR exposed to GW4064 was significantly reduced [44]. The ubiquitin–proteasome system and autophagy affect protein degradation [45], suggesting that FXR proteins may increase their stability and reduce degradation by binding with agonists, so we still need to explore the effect of CDCA on the ubiquitination levels of FXR proteins.

Additionally, there are some shortcomings in this study. We observed some degree of liver damage in the acute aspirin model, manifested by the activation of inflammation and without effect on FXR pathway target genes (Figure A3A–C). As bile acids are stored in the gallbladder after being synthesized in the liver and transported directly to the duodenum [46], our results also suggested that the disorder of the bile acid composition may be a risk factor for digestive tract inflammation, and the liver has enzymes (bile acid-CoA: amino acid N-acyltransferase) that specifically regulate the formation of conjugated bile acids to increase the water solubility and stability [47]. CDCA not only significantly inhibited the level of the BAAT gene, but also reduced the proportion of binding bile acids induced by ASA, suggesting that the intestinal protective effect of CDCA may be related to the production of bile acids in the liver in addition to the activation of FXR in the intestine (Figure A3D). It will be the next research plan to find out whether there is a certain relationship between liver damage and intestinal damage. Additionally, the environment of the ileum in animals is more complex, bile acids are mainly reabsorbed in the ileum, and the intestinal flora affects the changes in the bile acids through bile salt hydrolase (BSH) and secondary bile acid production [27]. We need more research on the composition and proportion of bile acids in the ileum. Due to the lack of studies on the role of ET-1 in gastrointestinal adverse reactions by other types of NSAIDs, we cannot confirm whether ET-1 inhibitors can be effective in a wider field, which means that studies on the gastrointestinal injuries by other NSAIDs need to be carry out.

## 4. Materials and Methods

### 4.1. Chemicals and Reagents

Aspirin (A2093), CDCA (C9377), and OCA (SML3096) were obtained from Sigma-Aldrich, Saint Louis, MO, USA. Estradiol (HY-B1100) was obtained from MedChemExpress, Monmouth Junction, NJ, USA. The fecal occult blood test kit was purchased from Leagene, Austin, TX, USA. The total bile acid (TBA, E00-1-1), DAO (A088-2-1), and malondialdehyde (MDA, A003-1-2) were obtained from Nanjing Jiancheng Bioengineering Institute, Nanjing, China. Dr. Wen Xie (University of Pittsburgh, PA, USA) kindly provided the plasmids of tk-EcRE-Luc and FXR [48]. The pRL-TK control vector was purchased from Promega (Madison, WI, USA).

### 4.2. Biochemical Analysis and Histological Analysis

TBA, DAO, and MDA were measured following the instructions of manufacturers. Fresh duodenal tissue was fixed in 4% paraformaldehyde, then dehydrated in 10% paraformaldehyde 48 h later. After paraffin embedding, the tissue was cut into 4 μm slices. It was dewaxed and rehydrated, then stained with hematoxylin–eosin for observation.

### 4.3. Sample Extraction and Metabolomics Analysis

The methods of metabolite extraction from duodenal tissue and the running status of the LC–MS system were performed as previously described [49,50]. Five μL of the sample solution was added into the UPLC–ESI–QTOFMS system. In this study, the extraction and processing software and methods of the LC–MS raw data were carried out as previously described [51]. The principal component analysis (PCA) and orthogonal projection to latent structures discriminant analysis (OPLS-DA) were analyzed using SIMCA-P + 14.1 (Umetrics, San Jose, CA, USA). The human metabolome database (HMDB) was used for the preliminary metabolite identification. The compounds were identified by the retention time and MS/MS fragment of the standards.

### 4.4. Animal Experiment

All experimental procedures followed the Guide for the Care and Use of Experimental Animals and were approved by the Animal Care and Use Committee of West China Hospital, Sichuan University (20231127008). Male 6-week-old *Fxr*-null mice (C57BL/6 J background) were cared for as previously described [52]. Male 6-week-old C57BL/6 mice (Gempharmatech Co., Ltd., Nanjing, Jiangsu, China) weighing 20–25 g were maintained under a standard 12 h light/12 h dark cycle environment with free access to water and rodent chow (Double lion experimental animal feed technology Co., Ltd., Suzhou, China). The mice were acclimatized in a suitable animal holding facility for 7 days and were anesthetized with isoflurane before death.

To establish the model of acute intestinal injury with aspirin, the WT mice were stochastically divided into the following groups (n = 5): (1) control group and (2) ASA group. After fasting for 16 h, mice of the ASA group were intragastrically administered aspirin (300 mg/kg [22], 0.5% CMC-Na). Mice were injected with 2% Evans Blue solution through the tail vein half an hour before being killed. The samples of serum and duodenal and ileal tissues were taken 3 h later. About 1 cm of tissue was fixed in 4% paraformaldehyde, and the remanent tissues were stored in the freezer at −80 °C.

To determine the effect of FXR on intestinal injury, the WT male mice were stochastically divided into three groups (n = 5): (1) control group; (2) ASA group; and (3) ASA + CDCA. Mice in the ASA + CDCA group were administrated CDCA (15 mg/kg, 1% DMSO + 2% Tween 80 + 97% water) for 5 days. After being fasted for 16 h, groups (2) and (3) were administrated aspirin (300 mg/kg), and we collected the serum and duodenal and ileal tissues 3 h later. 

In addition, OCA was used as an FXR agonist to carry out another batch of animal experiments in this study: the WT male mice were stochastically divided into three groups (n = 5): (1) control group; (2) ASA group; and (3) ASA + OCA. Mice in the ASA + OCA group were administrated OCA (10 mg/kg, 1% DMSO + 2% Tween 80 + 97% water) for 5 days. After being fasted for 16 h, groups (2) and (3) were administrated aspirin (300 mg/kg), and we collected the serum and duodenal and ileal tissues 3 h later.

*Fxr*-null (*Fxr^−/−^*) male mice were used to confirm the role of FXR in intestinal injury, and the *Fxr*-null (*Fxr^−/−^*) mice were stochastically divided into three groups (n = 5): (1) control group; (2) ASA group; and (3) ASA + CDCA. Mice in the ASA + CDCA group were administrated CDCA (15 mg/kg) for 5 days. After being fasted for 16 h, groups (2) and (3) were administrated aspirin (300 mg/kg), and we collected the serum and duodenal and ileal tissues 3 h after treatment.

To determine the relationship between FXR and ET-1 in aspirin-induced intestinal injury, the WT male mice were stochastically divided into three groups (n = 5): (1) control group; (2) ASA group; and (3) ASA + estradiol. The mice in group (3) were intraperitoneal injected with estradiol, which is the inhibitor of ET-1 (15 mg/kg, 1% DMSO + 99% corn oil), for 4 days. After being fasted for 16 h, groups (2) and (3) were administrated aspirin (300 mg/kg), and we collected the serum and duodenal and ileal tissues 3 h later.

To explore the role of FXR in the intestinal injury caused by long-term, low-dose aspirin, the WT male mice were stochastically divided into three groups (n = 5): (1) control group; (2) ASA group; and (3) ASA + CDCA. Mice in the ASA group were given aspirin (150 mg/kg [26], 0.5% CMC-Na) by intragastric administration for 14 days, and CDCA (15 mg/kg) was given to group (3) from the 8th day.

### 4.5. Detection of Gene Expression Levels

Trizol reagent (Vazyme, Nanjing, China) was applied to distill mRNA from about 20 mg of duodenum, ileum, and liver. Quantitative real-time PCR (qPCR) was carried out as previously described [51]; the primer sequences are listed in Appendix A Table A1. The mRNA levels of GAPDH were picked to normalize the target genes, and the thermal cycle conditions followed the scheme described previously [53]. The primer sequences used are provided in Appendix A Table A1. The stabilities of the house-keeping genes were checked using GeNorm and NormFinder in Appendix A Table A2.

### 4.6. Western Blotting

The protein extraction from the tissues and the Western blot analysis were performed as previously described [54]. The protein marker was the Tri-color Prestained Protein Marker (WJ103, Epizyme, Cambridge, MA, USA). The primary antibodies used were as follows: GAPDH (#5174, CST), p-JNK (#4668, CST), p-CFOS (#5348, CST), p-STAT3 (ab76315, Abcam, Cambridge, UK), t-JNK (#9252, CST), t-CFOS (#31254, CST), t-STAT3 (ab68153, Abcam), FXR (sc-25309, Santa, Frisco, TX, USA), SHP (ab232841, Abcam), and ET-1 (ab2786, Abcam). The secondary antibodies used were goat anti-mouse IgG HRP (HA1006, HUABIO, Woburn, MA, USA) and goat anti-rabbit IgG HRP (HA1001, HUABIO).

### 4.7. Statistical Analysis

Graphs were created in Prism 9.0.0 (GraphPad, La Jolla, CA, USA). The two-tailed student’s *t*-test for two sample group comparisons and a one-way ANOVA followed by Tukey’s post hoc test for multiple treatment comparison were carried out to evaluate the statistical significance. Data were expressed as the mean ± SD, and a *p* < 0.05 was considered statistically significant (* *p* < 0.05, ** *p* < 0.01, *** *p* < 0.001).

## 5. Conclusions

We found the mechanism of aspirin-induced intestinal injury through the inhibition of FXR and activation of ET-1 and confirmed the key regulatory role of FXR in drug-induced intestinal injury. Both FXR and ET-1 could be used as potential targets for the prevention and treatment of an aspirin-induced digestive tract injury, which provides a theoretical basis for improving the safety of clinical drug use. CDCA and OCA alleviated aspirin-induced intestinal injury by activating FXR. ET-1 inhibitors also exerted an effective anti-inflammatory and antioxidant role in protecting intestinal damage.

## Figures and Tables

**Figure 1 ijms-25-03424-f001:**
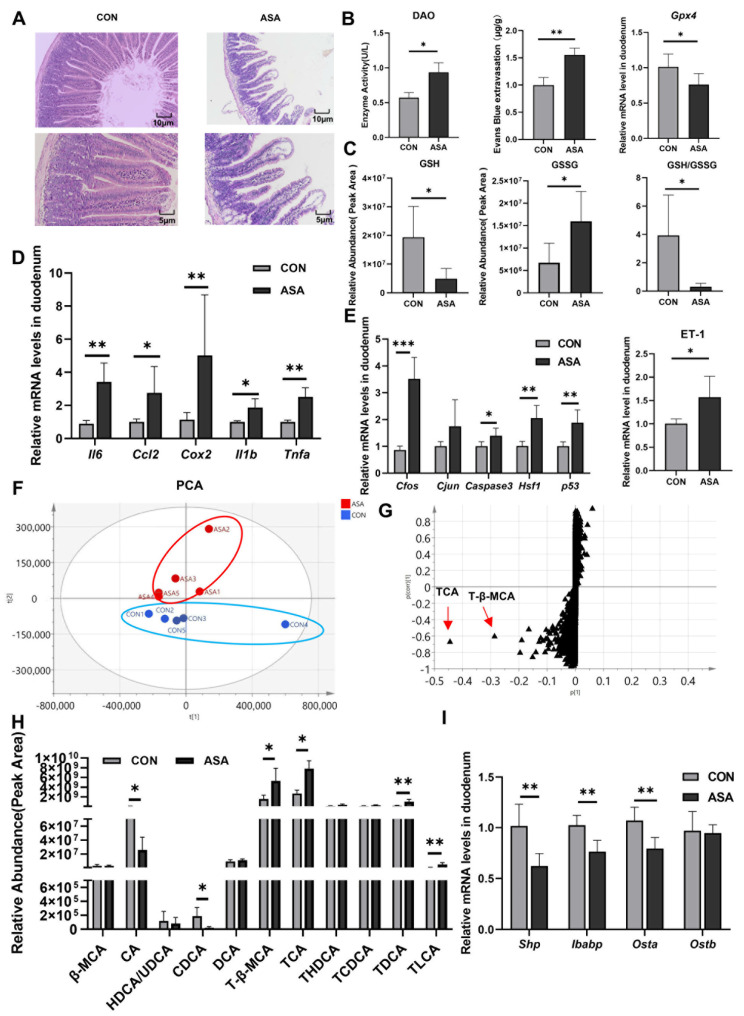
Disorder of bile acids and inhibition of FXR were found in aspirin-induced intestinal injury. (**A**) H&E of the duodenum. (**B**) DAO in the serum, Evans Blue extravasation, and *Gpx4* expression in the duodenum. (**C**) GSH, GSSG, and their ratio in the duodenum. (**D**,**E**) mRNA levels of inflammatory factor, JNK pathway genes, and ET-1. (**F**,**G**) PCA and OPLS-DA of duodenal metabolite. (**H**) Bile acid extraction from the duodenum. (**I**) FXR target gene levels. The data are expressed as the mean ± SD (* *p* < 0.05; ** *p* < 0.01; *** *p* < 0.001, compared with CON).

**Figure 2 ijms-25-03424-f002:**
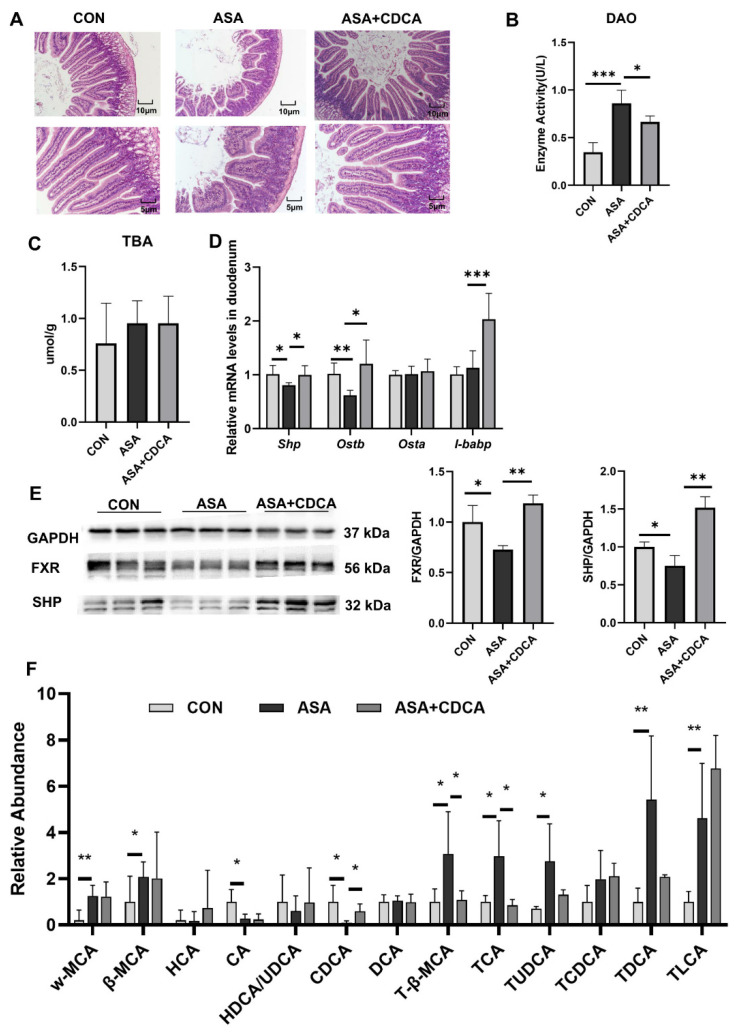
CDCA activation of FXR improves the disturbed proportion of bile acids in the gut. (**A**) H&E of the duodenum. (**B**) The enzyme activity of DAO in the serum. (**C**) TBA concentration in the duodenum. (**D**) FXR target gene expression levels in the duodenum. (**E**) Protein contents of FXR and SHP in the duodenum. (**F**) Targeted bile acid extraction from the duodenum. The data are expressed as the mean ± SD (* *p* < 0.05, compared with CON or ASA; ** *p* < 0.01, compared with CON or ASA; *** *p* < 0.001, compared with CON or ASA).

**Figure 3 ijms-25-03424-f003:**
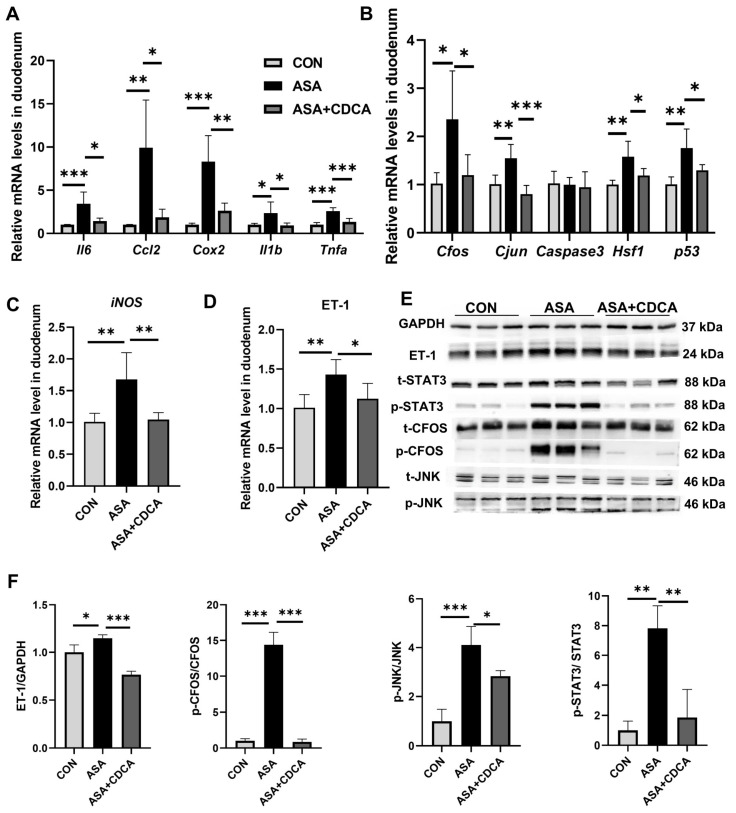
CDCA pretreatment mitigated aspirin-induced duodenum injury. (**A**,**B**) mRNA levels of inflammatory factors and JNK pathway genes in the duodenum. (**C**,**D**) mRNA levels of iNOS and ET-1 in the duodenum. (**E**,**F**) Protein contents of JNK pathway in the duodenum. The data are expressed as the mean ± SD (* *p* < 0.05, compared with CON or ASA; ** *p* < 0.01, compared with CON or ASA; *** *p* < 0.001, compared with CON or ASA).

**Figure 4 ijms-25-03424-f004:**
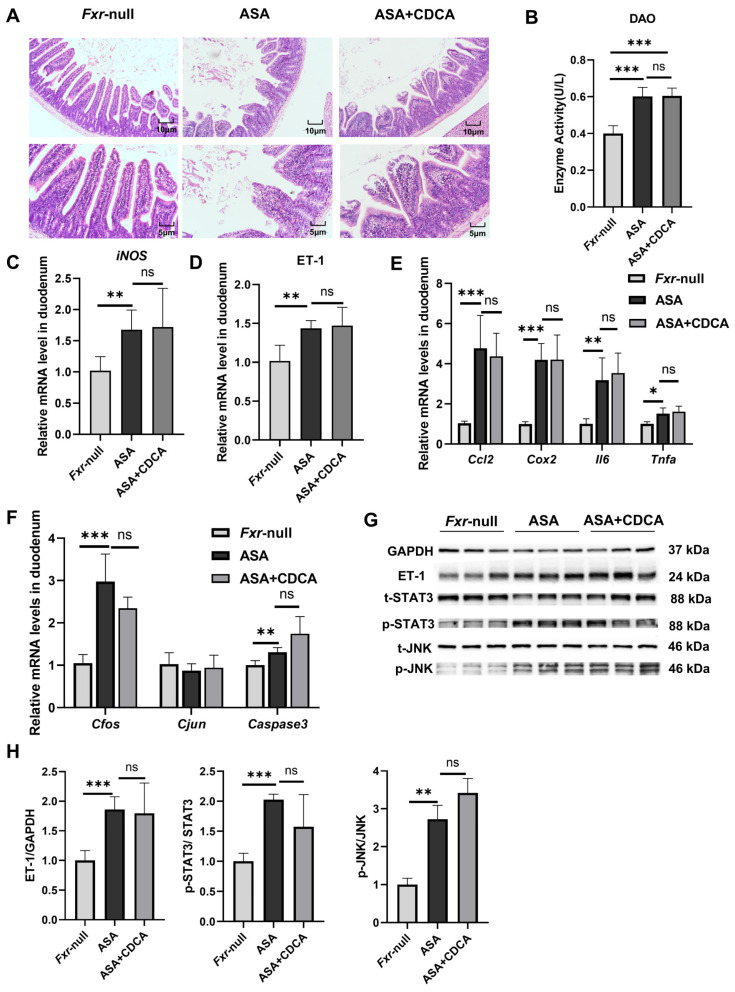
*Fxr^−/−^* blocked the protective role of CDCA in ASA-induced duodenum inflammation. (**A**) H&E of duodenums in *Fxr*-null mice after treatment with ASA and CDCA + ASA. (**B**) DAO concentration in the serum. (**C**–**F**) The expression levels of iNOS, ET-1, inflammatory factors, and JNK pathway genes in the duodenum. (**G**,**H**) The protein concentrations of ET-1 and JNK in the duodenums of *Fxr*-null mice. The data are expressed as the mean ± SD (ns: No statistical significance, compared with ASA; * *p* < 0.05, compared with CON or ASA; ** *p* < 0.01, compared with CON or ASA; *** *p* < 0.001, compared with CON or ASA).

**Figure 5 ijms-25-03424-f005:**
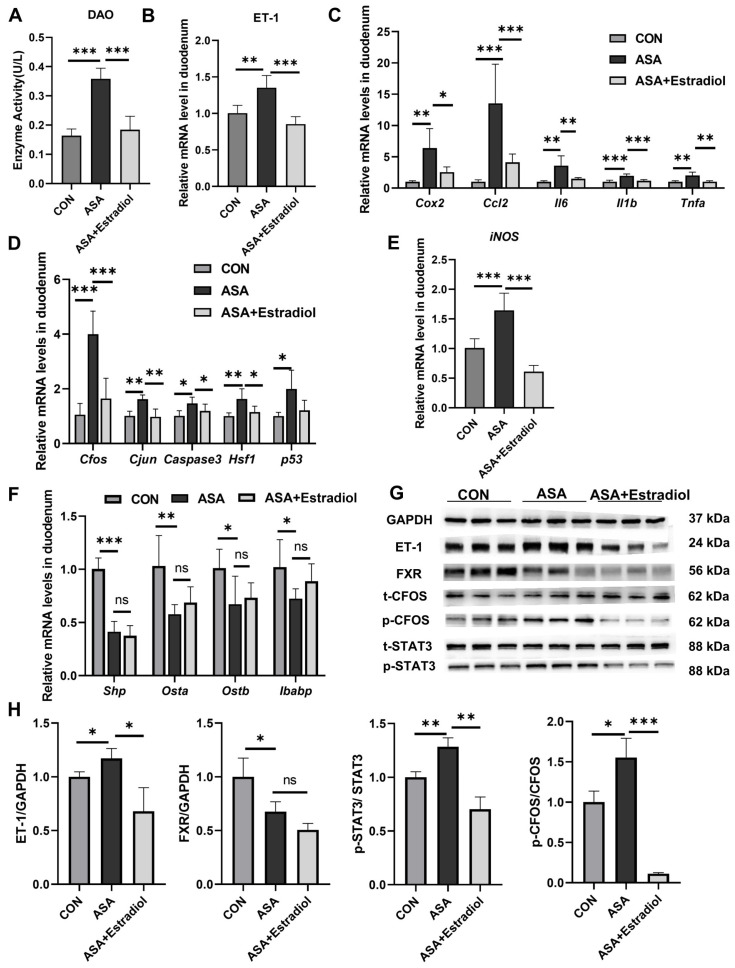
Estradiol relieved intestinal inflammation and JNK pathway overexpression by inhibiting the expression of ET-1. (**A**) DAO levels in the sera of mice treated with ASA and ASA + estradiol. (**B**) Estradiol inhibited ET-1 gene expression in the aspirin-induced intestine damage model. (**C**–**E**) Inflammatory cytokine, JNK pathway gene and iNOS gene levels in the duodenum. (**F**) FXR target gene expression in the duodenum. (**G**, **H**) Protein levels of ET-1, FXR, and JNK pathway genes in the duodenum. The data are expressed as the mean ± SD (* *p* < 0.05, compared with CON or ASA; ** *p* < 0.01, compared with CON or ASA; *** *p* < 0.001, compared with CON or ASA).

**Figure 6 ijms-25-03424-f006:**
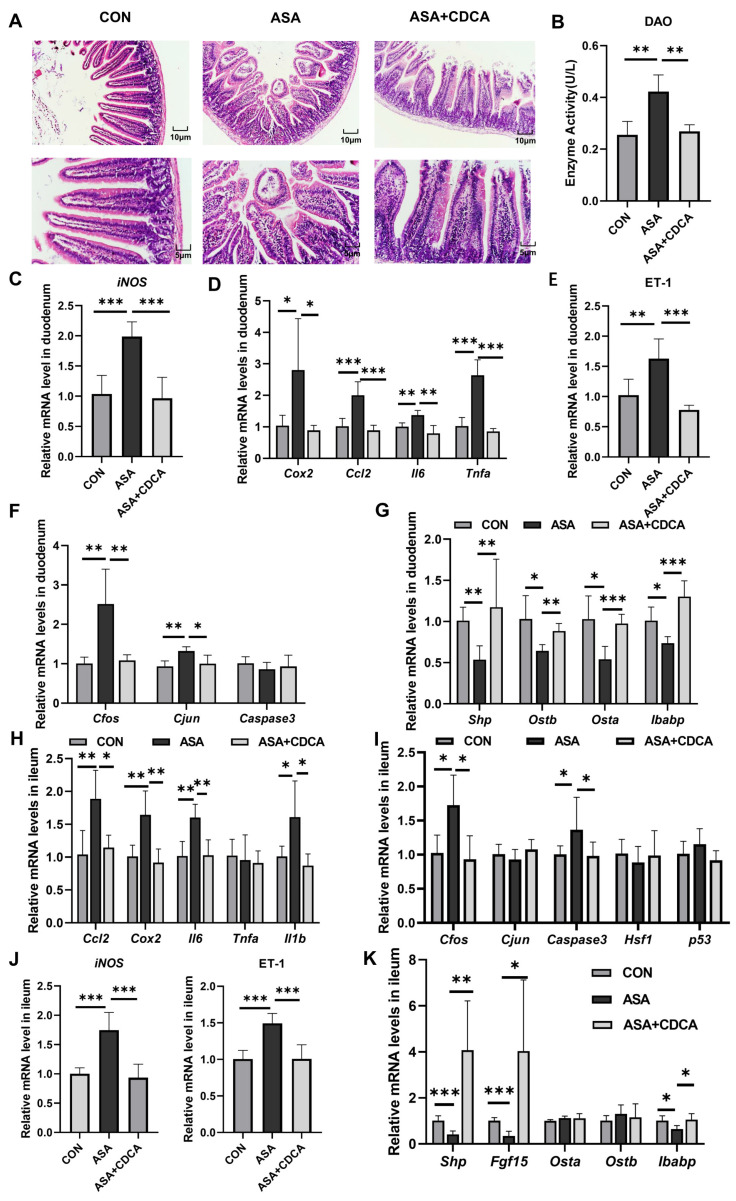
Activation of FXR improves chronic duodenal and ileal injuries caused by low-dose aspirin. (**A**) H&E staining of duodenums in mice with ASA and ASA + CDCA. (**B**) DAO concentration in the serum. (**C**–**F**) The gene expression levels of iNOS, inflammatory factors, ET-1, and JNK pathway genes in the duodenum. (**G**) Expression levels of different target genes of FXR in the duodenum. (**H**–**J**) Inflammatory factor, JNK pathway gene, iNOS, and ET-1 gene expressions in the ileum. (**K**) FXR target gene level in the ileum. The data are expressed as the mean ± SD (* *p* < 0.05, compared with CON or ASA; ** *p* < 0.01, compared with CON or ASA; *** *p* < 0.001, compared with CON or ASA).

## Data Availability

The data presented in this study are available on request from the corresponding author Fei Li (feili@wchscu.cn).

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
