# Peer review of "Aspirin Caused Intestinal Damage through FXR and ET-1 Signaling Pathways"

_ijms, 2024, doi:10.3390/ijms25063424_

Round 1

Reviewer 1 Report

Comments and Suggestions for Authors

In this manuscript Authors explore the mechanisms whereby an acute exposure to a large dose or a chronic exposure to a lower dose of aspirin cause gastrointestinal adverse reactions. They found that aspirin interferes with bile acid metabolism and inhibits FXR signaling pathways, while activates endothelin 1(ET-1) signaling pathway, inducing inflammation and tissue damage. The use of agonists of FXR and ET-1 inhibitor  show protective effects. In Fxr knockout mice Authors demonstrated a relationship between ET-1 and FXR. Authors conclude that FXR and ET-1 pathways have a key role in aspirin-induced intestinal  injury, and that activating FXR and inhibiting  ET-1 could represent new strategies for alleviating aspirin gastrointestinal injury.

Overall the paper is interesting, experimental design is solid and conclusion are well supported by results. 

I have only some comments :

-       How Authors chose aspirin concentrations?

-       In Appendix Fig.2G,H FXR expression after treatment with aspirin and aspirin+OCA, does not appear  altered in western blot,  contrarily  densitometric analysis shows significant alterations. Authors should explain this result.

-       In the chronic mice model, inhibition of the FXR pathway is shown in Fig 6.  Did Authors analyse also reduction of FXR expression?  

-       In fxr-null mice a western blot demonstrating the absence of FXR expression should be shown.

Author Response

Reviewer #1

In this manuscript Authors explore the mechanisms whereby an acute exposure to a large dose or a chronic exposure to a lower dose of aspirin cause gastrointestinal adverse reactions. They found that aspirin interferes with bile acid metabolism and inhibits FXR signaling pathways, while activates endothelin 1(ET-1) signaling pathway, inducing inflammation and tissue damage. The use of agonists of FXR and ET-1 inhibitor show protective effects. In Fxr knockout mice Authors demonstrated a relationship between ET-1 and FXR. Authors conclude that FXR and ET-1 pathways have a key role in aspirin-induced intestinal injury, and that activating FXR and inhibiting ET-1 could represent new strategies for alleviating aspirin gastrointestinal injury.

Overall the paper is interesting, experimental design is solid and conclusion are well supported by results. 

I have only some comments:

Comment 1: How Authors chose aspirin concentrations?

Reply 1: The dosage of aspirin (300 mg/kg) for acute injury was according to the literature [1-3] to cause obvious gastrointestinal damage. According to the literature on the use of low-dose aspirin to induce pancreatitis, and to save time and cost, we increased the dose to three times [4].

Comment 2: In Appendix Fig.2G, H FXR expression after treatment with aspirin and aspirin + OCA, does not appear altered in western blot, contrarily densitometric analysis shows significant alterations. Authors should explain this result.

Reply 2: The content of GAPDH protein in ASA + OCA group was slightly lower than that in CON and ASA groups, and the content of FXR protein was only slightly increased so that it doesn’t seem to change obviously.

Comment 3: In the chronic mice model, inhibition of the FXR pathway is shown in Fig 6.  Did Authors analyse also reduction of FXR expression?

Reply 3: We analyzed the changes of FXR protein expression in duodenum and ileum in the chronic model, which has been added to the attachment of the article in Appendix Figure 4D, E. And described them in " 2.5. CDCA improved aspirin-induced chronic duodenal and ileum damage through FXR and ET-1 pathway "(Line 225-227). FXR expression significantly decreased in ileum in ASA group, and CDCA could effectively activate FXR in duodenum and ileum.

(D-E) FXR protein expression in duodenum (D) and ileum (E).

Comment 4: In fxr-null mice a western blot demonstrating the absence of FXR expression should be shown.

Reply 4: Western Blot result of FXR null mice have been added to the manuscript in Appendix Figure 4A.

  • Genotype identification and protein expression of FXR knockout mice.

Reference:

  1. Dai, M.; Peng, W.; Lin, L.; Wu, Z.E.; Zhang, T.; Zhao, Q.; Cheng, Y.; Lin, Q.; Zhang, B.; Liu, A.; et al. Celastrol as an intestinal FXR inhibitor triggers tripolide-induced intestinal bleeding: Underlying mechanism of gastrointestinal injury induced by Tripterygium wilfordii. Phytomedicine : international journal of phytotherapy and phytopharmacology 2023, 121, 155054.
  2. Du, J.; Li, X.H.; Zhang, W.; Yang, Y.M.; Wu, Y.H.; Li, W.Q.; Peng, J.; Li, Y.J. Involvement of glutamate-cystine/glutamate transporter system in aspirin-induced acute gastric mucosa injury. Biochemical and biophysical research communications 2014, 450, 135-141.
  3. Chen, T.; Bao, S.; Chen, J.; Zhang, J.; Wei, H.; Hu, X.; Liang, Y.; Li, J.; Yan, S. Xiaojianzhong decoction attenuates aspirin-induced gastric mucosal injury via the PI3K/AKT/mTOR/ULK1 and AMPK/ULK1 pathways. Pharmaceutical biology 2023, 61, 1234-1248.
  4. Akyazi, I.; Eraslan, E.; Gülçubuk, A.; Ekiz, E.E.; Cırakli, Z.L.; Haktanir, D.; Bala, D.A.; Ozkurt, M.; Matur, E.; Ozcan, M. Long-term aspirin pretreatment in the prevention of cerulein-induced acute pancreatitis in rats. World journal of gastroenterology 2013, 19, 2894-2903.

Reviewer 2 Report

Comments and Suggestions for Authors

The work by Lin et al. explores the roles of the endothelin-1 (ET-1) and farnesoid X receptor (FXR) signaling pathways in the processes behind aspirin-induced intestinal damage. The authors show that aspirin administration results in a disturbance of the bile acid pool and inhibition of FXR in the colon using a combination of metabolomics, gene expression studies, and animal models. Due to the upregulation of ET-1 brought on by this suppression of FXR, intestinal damage, oxidative stress, and inflammation are consequently brought on.

Few major comments:

- The research uses a thorough and well-planned experimental strategy, combining a number of methods including gene expression analysis, metabolomics, and animal models. The employment of Fxr-knockout and wild-type mice, in conjunction with various agonists and inhibitors, offers a strong foundation for examining the function of FXR and ET-1 pathways in aspirin-induced intestinal damage. The authors should, however, go into further detail about the statistical analysis that was done and the standards that were applied to determine significance.

- The work uncovers a novel mechanism by which aspirin causes intestinal damage by activating the ET-1 signaling pathway and inhibiting FXR, which in turn causes inflammation and oxidative stress. The results underscore the potential of focusing on FXR and ET-1 as therapeutic approaches to alleviate gastrointestinal unpleasant responses caused by aspirin, an important clinical consideration. In the discussion and conclusions sections, the writers may underline even more how new and potentially useful their discoveries are for translation.

- Understanding the results is made easier by the well-organized results section and the understandable and informative figures and figure legends. In the discussion section, the writers offer a thorough analysis of the findings within the framework of the body of current literature. But some of the findings, especially the ones that deal with pathway analysis and protein expression, might benefit from more thorough justification.

- The authors should have been more open about any potential restrictions or warnings associated with their experimental methods. Though it could be expanded upon, the discussion of potential directions and consequences for further research is informative. It would improve the manuscript to address competing theories or contradicting findings from other research in the discussion.

Author Response

The work by Lin et al. explores the roles of the endothelin-1 (ET-1) and farnesoid X receptor (FXR) signaling pathways in the processes behind aspirin-induced intestinal damage. The authors show that aspirin administration results in a disturbance of the bile acid pool and inhibition of FXR in the colon using a combination of metabolomics, gene expression studies, and animal models. Due to the upregulation of ET-1 brought on by this suppression of FXR, intestinal damage, oxidative stress, and inflammation are consequently brought on.

Few major comments:

Commet 1: The research uses a thorough and well-planned experimental strategy, combining a number of methods including gene expression analysis, metabolomics, and animal models. The employment of Fxr-knockout and wild-type mice, in conjunction with various agonists and inhibitors, offers a strong foundation for examining the function of FXR and ET-1 pathways in aspirin-induced intestinal damage. The authors should, however, go into further detail about the statistical analysis that was done and the standards that were applied to determine significance.

Reply 1: We have added the method of statistical analysis and the standards applied to determine significance to the manuscript (Line 433-437).

“Graphs were conducted in Prism 9.0.0 (GraphPad, La Jolla, CA). Two-tailed stu-dent's t-test for two sample groups comparison and one-way ANOVA followed by Tukey's post-hoc test for multiple treatment comparison were carried out to evaluate statistical significance. Data were expressed as mean ± SD and a P < 0.05 was consid-ered statistically significant (*P < 0.05, **P < 0.01, ***P < 0.001).”

Commet 2: The work uncovers a novel mechanism by which aspirin causes intestinal damage by activating the ET-1 signaling pathway and inhibiting FXR, which in turn causes inflammation and oxidative stress. The results underscore the potential of focusing on FXR and ET-1 as therapeutic approaches to alleviate gastrointestinal unpleasant responses caused by aspirin, an important clinical consideration. In the discussion and conclusions sections, the writers may underline even more how new and potentially useful their discoveries are for translation.

Reply 2: In our manuscript, we highlighted the prevalence of ET-1 overactivation in NSAIDs and the novelty of its use as a direct target for the prevention and treatment of intestinal injury (Line 290-298).

“The expression level of ET-1 increased by 3.1 times in indomethacin gastric mucosal injury [1], and the correlation between ET-1 overexpression and JNK and inflammatory activation after aspirin administration has been further confirmed in our results, suggesting the reliability of ET-1 as a reference factor for the existence of NSAIDs intestinal injury. Few researches have used ET-1 as a direct target to alleviate gastrointestinal mucosal injury. Our study indicated that ET-1 may be a potential therapeutic target for gastrointestinal adverse reactions of NSAIDs, and its expression could be regulated by directly inhibiting or activating FXR to achieve prevention and treatment purposes.”

As for the role of FXR, previous studies have proved that activation of FXR can reduce the digestive tract injury caused by NSAIDs[2], and we already emphasized this point in the discussion section.

Commet 3: Understanding the results is made easier by the well-organized results section and the understandable and informative figures and figure legends. In the discussion section, the writers offer a thorough analysis of the findings within the framework of the body of current literature. But some of the findings, especially the ones that deal with pathway analysis and protein expression, might benefit from more thorough justification.

Reply 3: We added the following to the discussion (Line 267-273).

“Additionally, the overexpression of ET-1 was related to the phosphorylation level of JNK [3], which induced phosphorylation of JNK and c-Jun in time-dependent manners, and inhibition of JNK can negatively regulate ET-1 transcription level as well[4]. Meanwhile, ET-1 stimulated the activation of c-Jun/activator protein 1 (AP-1) through the Gq/i protein-coupled ETB receptor, then bonded and activated COX2 promoter, upregulated the transcription level of COX2, and induced inflammation of cerebral in microvascular endothelial cells [5]”, which make the relationship between ET-1 and JNK, inflammation regulation more clear.

Comment 4: The authors should have been more open about any potential restrictions or warnings associated with their experimental methods. Though it could be expanded upon, the discussion of potential directions and consequences for further research is informative. It would improve the manuscript to address competing theories or contradicting findings from other research in the discussion.

Reply 4: We added the limitations and further research direction of the study in the discussion section (Line 331-338).

“Besides, the environment of the ileum in animals is more complex, bile acids are mainly reabsorbed in the ileum, and intestinal flora affects the changes of bile acids through bile salt hydrolase (BSH) and the secondary bile acid production[6]. We need more research on the composition and proportion of bile acids in the ileum. Due to the lack of studies on the role of ET-1 in gastrointestinal adverse reactions of other types of NSAIDs, we cannot confirm whether ET-1 inhibitors can be effective in a wider field, which means the gastrointestinal injury of other NSAIDs need to be carry out”

Reference

  1. Slomiany, B.L.; Piotrowski, J.; Slomiany, A. Role of endothelin-1 and constitutive nitric oxide synthase in gastric mucosal resistance to indomethacin injury: effect of antiulcer agents. Scandinavian journal of gastroenterology 1999, 34, 459-464.
  2. Fiorucci, S.; Mencarelli, A.; Cipriani, S.; Renga, B.; Palladino, G.; Santucci, L.; Distrutti, E. Activation of the farnesoid-X receptor protects against gastrointestinal injury caused by non-steroidal anti-inflammatory drugs in mice. British journal of pharmacology 2011, 164, 1929-1938.
  3. Shi-Wen, X.; Rodríguez-Pascual, F.; Lamas, S.; Holmes, A.; Howat, S.; Pearson, J.D.; Dashwood, M.R.; du Bois, R.M.; Denton, C.P.; Black, C.M.; et al. Constitutive ALK5-independent c-Jun N-terminal kinase activation contributes to endothelin-1 overexpression in pulmonary fibrosis: evidence of an autocrine endothelin loop operating through the endothelin A and B receptors. Molecular and cellular biology 2006, 26, 5518-5527.
  4. Aktar, M.K.; Kido-Nakahara, M.; Furue, M.; Nakahara, T. Mutual upregulation of endothelin-1 and IL-25 in atopic dermatitis. Allergy 2015, 70, 846-854.
  5. Hsieh, H.L.; Lin, C.C.; Chan, H.J.; Yang, C.M.; Yang, C.M. c-Src-dependent EGF receptor transactivation contributes to ET-1-induced COX-2 expression in brain microvascular endothelial cells. Journal of neuroinflammation 2012, 9, 152.
  6. Huang, F.; Zheng, X.; Ma, X.; Jiang, R.; Zhou, W.; Zhou, S.; Zhang, Y.; Lei, S.; Wang, S.; Kuang, J.; et al. Theabrownin from Pu-erh tea attenuates hypercholesterolemia via modulation of gut microbiota and bile acid metabolism. Nature communications 2019, 10, 4971.